# Discovering the compositional structure of vector representations with Role Learning Networks

## Abstract

Neural networks are able to perform tasks that rely on compositional structure even though they lack obvious mechanisms for representing this structure. To analyze the internal representations that enable such success, we propose ROLE, a technique that detects whether these representations implicitly encode symbolic structure. ROLE learns to approximate the representations of a target encoder $\mathcal{E}$ by learning a symbolic constituent structure and an embedding of that structure into $\mathcal{E}$'s representational vector space. The constituents of the approximating symbol structure are defined by structural positions — roles — that can be filled by symbols. We show that when $\mathcal{E}$ is constructed to explicitly embed a particular type of structure (string or tree), ROLE successfully extracts the ground-truth roles defining that structure. We then analyze a GRU seq2seq network trained to perform a more complex compositional task (SCAN), where there is no ground truth role scheme available. For this model, ROLE successfully discovers an interpretable symbolic structure that the model implicitly uses to perform the SCAN task, providing a comprehensive account of the representations that drive the behavior of a frequently-used but hard-to-interpret type of model. We verify the causal importance of the discovered symbolic structure by showing that, when we systematically manipulate hidden embeddings based on this symbolic structure, the model's resulting output is changed in the way predicted by our analysis. Finally, we use ROLE to explore whether popular sentence embedding models are capturing compositional structure and find evidence that they are not; we conclude by suggesting how insights from ROLE can be used to impart new inductive biases to improve the compositional abilities of such models.

## 1 Overview

Certain AI tasks consist in computing a function $\varphi$ that is governed by strict rules: e.g., if $\varphi$ is the function mapping a mathematical expression to its value (e.g., mapping '$19-2*7$' to 5), then $\varphi$ obeys the rule that $\varphi(x+y) = \texttt{sum}(\varphi(x), \varphi(y))$ for any expressions $x$ and $y$. This rule is **compositional**: the output of a structure (here, $x+y$) is a function of the outputs of the structure's constituents (here, $x$ and $y$). The rule can be stated with full generality once the input is assigned a **symbolic structure** giving its decomposition into constituents. For a **fully-compositional** task, completely determined by compositional rules, an AI system that can assign appropriate symbolic structures to inputs and apply appropriate compositional rules to these structures will display full **systematic generalization**: it will correctly process arbitrary novel combinations of familiar constituents. This is a core capability of symbolic AI systems. Other tasks, including most natural language tasks such as machine translation, are only partially characterizable by compositional rules because natural language is only partially compositional in nature. For example, if $\varphi$ is the function that assigns meanings to English adjectives, it generally obeys the rule that $\varphi(\texttt{in-} + x) = \texttt{not } \varphi(x)$, (e.g., $\varphi(\texttt{inoffensive}) = \texttt{not } \varphi(\texttt{offensive})$), yet there are exceptions: $\varphi(\texttt{inflammable}) = \varphi(\texttt{flammable})$. On these "**partially-compositional**" AI tasks, this strategy of compositional analysis has demonstrated considerable, but limited, generalization capabilities.

Deep learning research has shown that Neural Networks (NNs) can display remarkable degrees of combinatorial generalization, often surpassing symbolic AI systems for partially-compositional tasks

(Wu et al., 2016), and exhibit good generalization (although generally falling short of symbolic AI systems) on fully-compositional tasks (Lake & Baroni, 2018; McCoy et al., 2019a). Given that standard NNs have no obvious mechanisms for representing symbolic structures, parsing inputs into such structures, nor applying compositional symbolic rules to them, this success raises the question we address in this paper: *How do NNs achieve such strong generalization on partially-compositional tasks, and good performance on fully-compositional tasks?*

An important step towards answering this question was reported in McCoy et al. (2019a), which showed that when trained on highly compositional tasks, standard NNs learned representations that were well approximated by symbolic structures (Sec. 2). Processing in these NNs assigns such representations to inputs and generates outputs that are governed by compositional rules stated over those representations. We refer to the networks to be analyzed as **target NNs**, because we will propose a new type of NN (in Sec. 4) — the **Role Learner (ROLE)** — which is used to *analyze* the target network, after first discussing related analysis methods in Sec. 3. In contrast with the analysis model of McCoy et al. (2019a), which relies on a hand-specified hypothesis about the underlying structure, ROLE *automatically* learns a symbolic structure that best approximates the internal representation of the target network. Automating the discovery of structural hypotheses provides two advantages. First, ROLE achieves success at analyzing networks for which it is not clear what the underlying structure is. We show this in Sec. 6, where ROLE successfully uncovers the symbolic structures learned by a seq2seq RNN trained on the SCAN task (Lake & Baroni, 2018). Second, removing the need for hand-specified hypotheses allows the data to speak for itself, which simplifies the burden on the user, who only needs to provide input sequences and associated embeddings.

We first consider fully-compositional (hence synthetic) tasks: a simple string-manipulation task in Sec. 5, and the richer SCAN task, which has been the basis of previous work on combinatorial generalization in NNs, in Sec. 6. Discovering symbolic structure within a model enables us to perform precise alterations to the internal representations in order to produce desired combinatorial alterations in the output (Sec. 6.3). Then, in Sec. 7, we turn briefly to partially-compositional tasks in NLP. In Sec. 8 we consider how what we have learned about standard NNs can suggest new inductive biases to strengthen compositionality in NN learning.

## 2   NN EMBEDDING OF SYMBOL STRUCTURES

We build on McCoy et al. (2019a), which introduced the analysis task **DISCOVER (DISsecting COmpositionality in VEctor Representations)**: take a NN and, to the extent possible, find an explicitly-compositional approximation to its internal distributed representations. McCoy et al. (2019a) showed that, in GRU (Cho et al., 2014) encoder-decoder networks performing simple, fully-compositional string manipulations, the medial encoding (between encoder and decoder) could be extremely well approximated, up to a linear transformation, by **Tensor Product Representations (TPRs)** (Smolensky, 1990), which are explicitly-compositional vector embeddings of symbolic structures. To represent a string of symbols as a TPR, the symbols in the string 337 might be parsed into three constituents $\{3 : \mathrm{pos}1, 7 : \mathrm{pos}3, 3 : \mathrm{pos}2\}$, where $\mathrm{pos}n$ is the role of $n^{\mathrm{th}}$ position from the left edge of the string; other role schemes are also possible, such as roles denoting right-to-left position: $\{3 : \mathrm{third\text{-}to\text{-}last}, 3 : \mathrm{second\text{-}to\text{-}last}, 7 : \mathrm{last}\}$. The embedding of a constituent $7 : \mathrm{pos}3$ is $\mathbf{e}(7 : \mathrm{pos}3) = \mathbf{e}_{\mathrm{F}}(7) \otimes \mathbf{e}_{\mathrm{R}}(\mathrm{pos}3)$, where $\mathbf{e}_{\mathrm{R}}, \mathbf{e}_{\mathrm{F}}$ are respectively a vector embedding of the roles and a vector embedding of the **fillers** of those roles: the digits. The embedding of the whole string is the sum of the embeddings of its constituents. In general, for a symbol structure $\mathtt{S}$ with roles $\{r_k\}$ that are respectively filled by the symbols $\{\mathtt{f}_k\}$, $\mathbf{e}_{\mathrm{TPR}}(\mathtt{S}) = \sum_k \mathbf{e}_{\mathrm{F}}(\mathtt{f}_k) \otimes \mathbf{e}_{\mathrm{R}}(r_k)$.

The example above used the **role scheme** of linear position from the left edge, but other role schemes are possible, such as linear position from the right edge or position in a tree. McCoy et al. (2019a) showed that — for a given seq2seq architecture learning a given string-mapping task — there exists a highly accurate TPR approximation of the medial encoding, given an appropriate pre-defined role scheme. The main technical contribution of the present paper is the Role Learner (ROLE) model, a RNN network that learns its own role scheme to optimize the fit of a TPR approximation to a given set of internal representations in a pre-trained target NN. This makes the DISCOVER framework more general by removing the need for human-generated hypotheses as to the role schemes the network might be implementing. Learned role schemes, we will see in Sec. 6.1, can enable good TPR approximation of networks for which human-generated role schemes fail.

## 3  RELATED WORK

This work falls within the larger paradigm of using analysis techniques to interpret NNs (see Belinkov & Glass (2019) for a recent survey), often including a focus on compositional structure (Hupkes et al., 2019; 2018; Lake & Baroni, 2018; Hewitt & Manning, 2019). Two of the most popular analysis techniques are the behavioral and probing approaches. In the behavioral approach, a model is evaluated on a set of examples carefully chosen to require competence in particular linguistic phenomena (Marvin & Linzen, 2018; Wang et al., 2018; Dasgupta et al., 2019; Poliak et al., 2018; Linzen et al., 2016; McCoy et al., 2019b). This technique can illuminate behavioral shortcomings but says little about how the internal representations are structured, treating the model as a black box.

In the probing approach, an auxiliary classifier is trained to classify the model's internal representations based on some linguistically-relevant distinction (Adi et al., 2017; Conneau et al., 2018; Conneau & Kiela, 2018; Belinkov et al., 2017; Blevins et al., 2018; Peters et al., 2018; Tenney et al., 2019). In contrast with the behavioral approach, the probing approach tests whether some particular information is present in the model's encodings, but it says little about whether this information is actually used by the model. Indeed, in at least some cases models will fail despite having the necessary information to succeed in their representations, showing that the ability of a classifier to extract that information does not mean that the model is using it (Vanmassenhove et al., 2017).

DISCOVER bridges the gap between representation and behavior: It reveals not only what information is encoded in the representation, but also shows how that information is causally implicated in the model's behavior. Moreover, it provides a much more comprehensive window into the representation than the probing approach does; while probing extracts particular types of information from a representation (e.g., "does this representation distinguish between active and passive sentences?"), DISCOVER exhaustively decomposes the model's representational space. In this regard, DISCOVER is most closely related to the approaches of Andreas (2019), Chrupała & Alishahi (2019), and Abnar et al. (2019), who also propose methods for discovering a complete symbolic characterization of a set of vector representations, and Omlin & Giles (1996) and Weiss et al. (2018), which also seek to extract more interpretable symbolic models that approximate neural network behavior.

## 4  THE ROLE LEARNER (ROLE) MODEL

ROLE[1] produces a vector-space embedding of an input string of $T$ symbols $\mathtt{S} = \mathtt{s}_1\mathtt{s}_2\ldots\mathtt{s}_T$ by producing a TPR $\mathbf{T}(\mathtt{S})$ and then passing it through a linear transformation $\mathbb{W}$. ROLE is trained to approximate a pre-trained target string-encoder $\mathcal{E}$. Given a set of $N$ training strings $\{\mathtt{S}^{(1)}, \ldots, \mathtt{S}^{(N)}\}$, ROLE minimizes the total mean-squared error (MSE) between its output $\mathbb{W}\,\mathbf{T}(\mathtt{S}^{(i)})$ and $\mathcal{E}$'s corresponding output, $\mathcal{E}(\mathtt{S}^{(i)})$.

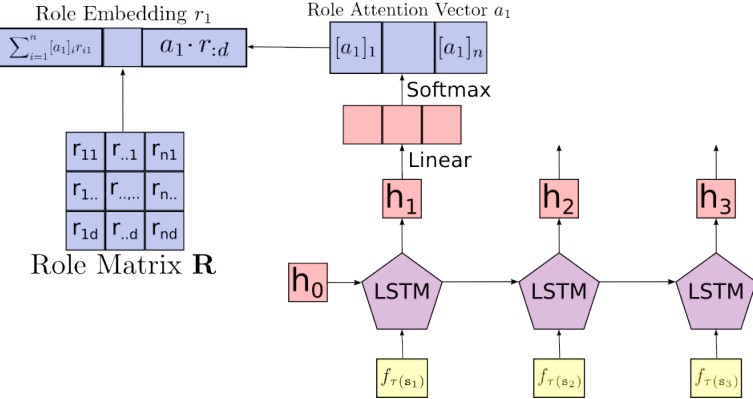

Figure 1: The role learning module. The role attention vector $a_t$ is encouraged to be one-hot through regularization; if $a_t$ were one-hot, the produced role embedding $r_t$ would correspond directly to one of the roles defined in the role matrix $\boldsymbol{R}$. The LSTM can be unidirectional or bidirectional.

---

[1]Code available at https://github.com/iclr2020-anonymous1/role-learner

ROLE is an extension of the Tensor-Product Encoder (TPE) introduced in McCoy et al. (2019a) (as the "Tensor Product Decomposition Network"), which produces a linearly-transformed TPR given a string of symbols and pre-assigned role labels for each symbol (see Appendix A.1 for details). Crucially, ROLE is not *given* role labels for the input symbols, but *learns to compute* them. More precisely, it learns a dictionary of $n_{\mathrm{R}}$ $d_{\mathrm{R}}$-dimensional role-embedding vectors, $\boldsymbol{R} \in \mathbb{R}^{d_{\mathrm{R}} \times n_{\mathrm{R}}}$, and, for each input symbol $\mathbf{s}_t$, computes a soft-attention vector $\boldsymbol{a}_t$ over these role vectors: the role vector assigned to $\mathbf{s}_t$ is then the attention-weighted linear combination of role vectors, $\boldsymbol{r}_t = \boldsymbol{R}\,\boldsymbol{a}_t$. ROLE simultaneously learns a dictionary of $n_{\mathrm{F}}$ $d_{\mathrm{F}}$-dimensional symbol-embedding filler vectors $\boldsymbol{F} \in \mathbb{R}^{d_{\mathrm{F}} \times n_{\mathrm{F}}}$, the $\phi^{\mathrm{th}}$ column of which is $\boldsymbol{f}_\phi$, the embedding of symbol type $\phi$; $\phi \in 1, \ldots, n_{\mathrm{F}}$ where $n_{\mathrm{F}}$ is the size of the vocabulary of symbol types. The TPR generated by ROLE is thus $\mathbf{T}(\mathbf{S}) = \sum_{t=1}^{T} \boldsymbol{f}_{\tau(\mathbf{s}_t)} \otimes \boldsymbol{r}_t$, where $\tau(\mathbf{s}_t)$ is symbol $\mathbf{s}_t$'s type. Finally, ROLE learns a linear transformation $\mathbb{W}$ to map this TPR into $\mathbb{R}^d$, where $d$ is the dimension of the representations of the encoder $\mathcal{E}$ it is learning to approximate.

ROLE uses an LSTM (Hochreiter & Schmidhuber, 1997) to compute the role-assigning attention-vectors $\boldsymbol{a}_t$ from its learned embedding $\boldsymbol{F}$ of the input symbols $\mathbf{s}_t$: at each $t$, the hidden state of the LSTM passes through a linear layer and then a softmax to produce $\boldsymbol{a}_t$ (depicted in Figure 1).[2]

Since a TPR for a discrete symbol structure deploys a discrete set of roles specifying discrete structural positions, ideally a single role would be selected for each $\mathbf{s}_t$: $\boldsymbol{a}_t$ would be one-hot. ROLE training therefore deploys regularization to bias learning towards one-hot $\boldsymbol{a}_t$ vectors (based on the regularization proposed in Palangi et al. (2017), developed for the same purpose). See Appendix A.4 for the precise regularization terms that we used. Although the regularization can yield good results, performing continuous gradient descent on a nearly discrete symbolic landscape can cause networks to get stuck in local optima where the learned symbolic structure is not ideal. For this reason, the performance at convergence can vary appreciably across multiple runs.

It is important to note that, while we impose this regularization on ROLE, there is no explicit bias favoring discrete compositional representations in the *target encoder* $\mathcal{E}$: any such structure that ROLE finds hidden in the representations learned by $\mathcal{E}$ must be the result of biases implicit in the vanilla RNN-architecture of $\mathcal{E}$ when applied to its target task.

## 5    A SIMPLE FULLY-COMPOSITIONAL TASK

We first apply ROLE to two target Tensor Product Encoder (TPE) models which are fully compositional by design. Since we know what role scheme each target model deploys, we can test how well ROLE learns these ground-truth roles. The TPEs are trained on the fully compositional task of autoencoding sequences of digits.

We use two types of TPEs: one that uses a simple left-to-right role scheme (e.g., *first*, *second*, *third*) and one that uses a complex tree position role scheme (e.g., *left child of the root of the tree*, *right child of the left child of the left child of the root of the tree*), where the trees are generated from digit sequence inputs using a deterministic parsing algorithm (see Appendix A.2 for explanations and examples of the designed role schemes). The left-to-right TPE was paired with a unidirectional RNN decoder, while the tree-position TPE was paired with the tree-RNN decoder used by McCoy et al. (2019a). Both of these target models attained near-perfect performance on the autoencoding task (Table 1). Once the encoders are finished training, we extract the encoding for each sequence in the dataset and use this to train ROLE. See Appendices A.3 and A.5 for additional training details.

Table 1 reports the approximation performance of ROLE in two ways. **Substitution Accuracy** is the proportion of the items for which the decoder produces the correct output string when it is fed the ROLE approximation. The **V-Measure** (Rosenberg & Hirschberg, 2007) assesses the extent to which the clustering of the role vectors assigned by ROLE matches the ground truth role assignments.

The ROLE approximation of the left-to-right TPE attained perfect performance, with a substitution accuracy of 100% and a V-Measure of 1.0, indicating that the role scheme it learned perfectly matched the ground truth. On the significantly more complex case of tree position roles, ROLE achieves

---

[2]Let the $t^{\mathrm{th}}$ LSTM hidden state be $\boldsymbol{q}_t \in \mathbb{R}^H$; let the output-layer weight-matrix have rows $\boldsymbol{k}_\rho^\top \in \mathbb{R}^H$ and let the columns of $\boldsymbol{R}$ be $\boldsymbol{v}_\rho \in R^{d_{\mathrm{R}}}$, with $\rho = 1, \ldots, n_{\mathrm{R}}$. Then $\boldsymbol{r}_t = \boldsymbol{R}\,\boldsymbol{a}_t = \sum_{\rho=1}^{n_{\mathrm{R}}} \boldsymbol{v}_\rho \,\mathrm{softmax}(\boldsymbol{k}_\rho^\top \boldsymbol{q}_t)$: the result of query-key attention (e.g., Vaswani et al., 2017) with query $\boldsymbol{q}_t$ to a fixed external memory containing key-value pairs $\{(\boldsymbol{k}_\rho, \boldsymbol{v}_\rho)\}_{\rho=1}^{n_{\mathrm{R}}}$.

Table 1: ROLE mean substitution accuracy and and role scheme v-measure along with one standard deviation across three random initializations on models that are fully compositional by design (TPEs).

| Target model | Target model accuracy | ROLE substitution accuracy | ROLE V-Measure |
|---|---|---|---|
| Left-to-right TPE | 100% | 100% $\pm 0$ | 1.0 $\pm 0$ |
| Tree-position TPE | 98.62% | 98.61 $\pm .05\%$ | 0.815 $\pm .01$ |

essentially the same accuracy as the target encoder $\mathcal{E}$ and has considerable success at recovering the ground truth roles for the vectors it was analyzing. These results show that, when a target model has a known fully compositional structure, ROLE can successfully find that structure.

## 6 THE SCAN TASK

We have established that ROLE can uncover the compositional structure used by a model that is compositional by design. But, returning to our central question from Sec. 1, how can models *without* explicit compositional structure (namely, standard RNNs) still be as successful at fully compositional tasks as fully compositional models? Our hypothesis is that, though these models have no *constraint* forcing them to be compositional, they still have the *ability* to implicitly learn compositional structure. To test this hypothesis, we apply ROLE to a standard RNN-based seq2seq (Sutskever et al., 2014) model trained on a fully compositional task. Because the RNN has no constraint forcing it to use TPRs, we do not know *a priori* whether there exists any solution that ROLE could learn; thus, if ROLE does learn anything it will be a significant empirical finding about how these RNNs operate.

We consider the SCAN task (Lake & Baroni, 2018), which was designed to test compositional generalization and systematicity. SCAN is a synthetic sequence-to-sequence mapping task, with an input sequence describing an action plan, e.g., `jump opposite left`, being mapped to a sequence of primitive actions, e.g., `TL TL JUMP` (see Sec. 6.3 for a complex example). We use `TL` to abbreviate `TURN_LEFT`, sometimes written `LTURN`; similarly, we use `TR` for `TURN_RIGHT`. The SCAN mapping is defined by a complete set of compositional rules (Lake & Baroni, 2018, Supplementary Fig. 7).

### 6.1 THE COMPOSITIONAL STRUCTURE OF SCAN ENCODER REPRESENTATIONS

For our target SCAN encoder $\mathcal{E}$, we trained a standard GRU with one hidden layer of dimension 100 for 100,000 steps (batch-size 1) with a dropout of 0.1 on the simple train-test split. (The hidden dimension and dropout rate were determined by a limited hyper-parameter search; see Appendix A.6). $\mathcal{E}$ achieves 98.47% (full-string) accuracy on the test set. Thus $\mathcal{E}$ provides what we want: a standard RNN achieving near-perfect accuracy on a non-trivial fully compositional task.

After training, we extract the final hidden embedding from the encoder for each example in the training and test sets. These are the encodings we attempt to approximate as explicitly compositional TPRs. We provide ROLE with 50 roles to use as it wants (additional training information is in Appendix A.7). We evaluate the substitution accuracy that this learned role scheme provides in three ways. The **continuous** method tests ROLE in the same way as it was trained, with input symbol $s_t$ assigned role vector $r_t = R\,a_t$. The continuous method does not use a discrete set of role vectors because the weighted sum that generates $a_t$ allows for continuously-valued weights.The remaining two methods test the efficacy of a truly discrete set of role vectors. First, in the **snapped** method, $a_t$ is replaced at evaluation time by the one-hot vector $m_t$ singling out role $m_t = \arg\max(a_t)$: $r_t = R\,m_t$. This method serves the goal of enforcing the discreteness of roles, but it is expected to decrease performance because it tests ROLE in a different way than it was trained. Our final evaluation method, the **discrete** method, uses discrete roles without having such a train/test discrepancy; in this method, we use the one-hot vector $m_t$ to output roles for every symbol in the dataset and then train a TPE which does not learn roles but rather uses the one-hot vector $m_t$ as input during training. In this case, ROLE acts as an automatic data labeler, assigning a role to every input word.

For comparison, we also train TPEs using a variety of discrete hand-crafted role schemes: left-to-right (LTR), right-to-left (RTL), bidirectional (Bi), tree position, Wickelrole (Wickel), and bag-of-words (BOW) (additional information provided in Appendix A.2).

Table 2: Mean substitution accuracy for learned (bold) and pre-defined role schemes on SCAN across three random initializations. Standard deviation was below 1% for all schemes except for snapped. Substitution accuracy is measured by feeding ROLE's approximation to the target decoder.

| Continuous | Snapped | Discrete | LTR | RTL | Bi | Tree | Wickel | BOW |
|---|---|---|---|---|---|---|---|---|
| 94.83% | 81.71% ±7.28 | 92.44% | 6.68% | 6.96% | 10.72% | 4.31% | 44.00% | 4.52% |

The substitution accuracy from these different methods is shown in Table 2. All of the predefined role schemes provide poor approximations, none surpassing 44.12% accuracy. The role scheme learned by ROLE does significantly better than any of the predefined role schemes: when tested with the basic, continuous role-attention method, the accuracy is 94.12%. The success of ROLE tells us two things. First, it shows that the target model's compositional behavior relies on compositional internal representations: it was by no means guaranteed to be the case that ROLE would be successful here, so the fact that it is successful tells us that the encoder has learned compositional representations. Second, it adds further validation to the efficacy of ROLE, because it shows that it can be a useful analysis tool in cases of significantly greater complexity than the autoencoding task.

## 6.2 Interpreting the learned role scheme

Analyzing the roles assigned by ROLE to the sequences in the SCAN training set, we created a symbolic algorithm for predicting which role will be assigned to a given filler. This is described in Appendix A.8.1 and discussed at some length in Appendix A.8.2. Though the algorithm was created based only on sequences in the SCAN training set, it is equally successful at predicting which roles will be assigned to test sequences, exactly matching ROLE's predicted roles for 98.7% of sequences.

The details of this algorithm illuminate how the filler-role scheme encodes information relevant to the task. First, one of the initial facts that the decoder must determine is whether the sequence is a single command, a pair of commands connected by `and`, or a pair of commands connected by `after`; such a determination is crucial for knowing the basic structure of the output (how many actions to perform and in what order). We have found that role 30 is used for, and only for, the filler `and`, while role 17 is used in and only in sequences containing `after` (usually with `after` as the filler bound to role 17). Thus, the decoder can use these roles to tell which basic structure is in play: if role 30 is present, it is an `and` sequence; if role 17 is present, it is an `after` sequence; otherwise it is a single command.

Once the decoder has established the basic syntactic structure of the output, it must then fill in the particular actions. This can be accomplished using the remaining roles, which mainly encode absolute position within a command. For example, the last word of a command before `after` (e.g., `jump` **left** `after walk twice`) is always assigned role 8, while the last word of a command after `after` (e.g., `jump left after walk` **twice**) is always assigned role 46. Therefore, once the decoder knows (based on the presence of role 17) that it is dealing with an `after` sequence, it can check for the fillers bound to roles 8 and 46 to begin to figure out what the two subcommands surrounding `after` look like. The identity of the last word in a command is informative because that is where a cardinality (i.e., `twice` or `thrice`) appears if there is one. Thus, by checking what filler is at the end of a command, the model can learn whether there is a cardinality present and, if so, which one.

This description of how the decoding could take place does not necessarily match how it actually does take place; for example, it is likely that some of the steps we have described as occurring serially, for expository simplicity, actually occur in parallel. We leave for future work the question of which operations are actually being performed and how those operations are instantiated in an RNN.

## 6.3 Precision constituent-surgery on internal representations to produce desired outputs

The substitution-accuracy results above show that if the *entire* learned representation is replaced by ROLE's approximation, the output remains correct. But do the *individual words* in this TPR have the appropriate causal consequences when processed by the decoder?[3]

---

[3]Historically, this question has had considerable significance: the original compositionality challenge to neural network models of cognition by Fodor and colleagues (Fodor & Pylyshyn, 1988) insisted that constituents

```
run: 11 left: 36 twice: 8 after: 43 jump: 10 opposite: 17 right: 4 thrice: 46 →
    TR TR JUMP TR TR JUMP TR TR JUMP TL RUN TL RUN
− run: 11 + look: 11 →
    TR TR JUMP TR TR JUMP TR TR JUMP TL LOOK TL LOOK
− jump: 10 + walk: 10 →
    TR TR WALK TR TR WALK TR TR WALK TL LOOK TL LOOK
− left: 36 + right: 36 →
    TR TR WALK TR TR WALK TR TR WALK TR LOOK TR LOOK
− twice: 8 + thrice: 8 →
    TR TR WALK TR TR WALK TR TR WALK TR LOOK
    TR LOOK TR LOOK
− opposite: 17 + around: 17 →
    TR WALK TR WALK TR WALK TR WALK TR WALK TR WALK
    TR WALK TR WALK TR WALK TR WALK TR WALK TR WALK
    TR LOOK TR LOOK TR LOOK
```

Accuracy by number of substitutions

Figure 2: Left: Example of successive constituent surgeries. The roles assigned to the input symbols are indicated in the first line (e.g., `run` was assigned role 11). Altered output symbols are in blue. The model produces the correct outputs for all cases shown here. Right: Mean constituent-surgery accuracy across three runs. Standard deviation is below 1% for each number of substitutions.

To address this causal question (Pearl, 2000), we actively intervene on the constituent structure of the internal representations by replacing one constituent with another syntactically equivalent one[4], and see whether this produces the expected change in the output of the decoder. We take the encoding generated by the RNN encoder $\mathcal{E}$ for an input such as `jump opposite left`, subtract the vector embedding of the `opposite` constituent, add the embedding of the `around` constituent, and see whether this causes the output to change from the correct output for `jump opposite left` (`TL TL JUMP`) to the correct output for `jump around left` (`TL JUMP TL JUMP TL JUMP TL JUMP`). The roles in these constituents are determined by the algorithm of Appendix A.8. If changing a word leads other roles in the sequence to change (according to the algorithm), we update the encoding with those new roles as well. Such surgery can be viewed as based in a more general extension of the analogy approach used by Mikolov et al. (2013) for analysis of word embeddings. An example of applying a sequence of five such constituent surgeries to a sequence are shown in Figure 2 (left).

## 7 PARTIALLY-COMPOSITIONAL NLP TASKS

The previous sections explored fully-compositional tasks where there is a strong signal for compositionality. In this section, we explore whether the representations of NNs trained on tasks that are only partially-compositional also capture compositional structure. Partially-compositional tasks are especially challenging to model because a fully-compositional model may enforce compositionality too strictly to handle the non-compositional aspects of the task, while a model without a compositional bias may not learn any sort of compositionality from the weak cues in the training set.

We test four sentence encoding models for compositionality: InferSent (Conneau et al., 2017), Skip-thought (Kiros et al., 2015), Stanford Sentiment Model (SST) (Socher et al., 2013), and SPINN (Bowman et al., 2016). For each of these models, we extract the encodings for the SNLI premise sentences (Bowman et al., 2015). We use the extracted embeddings to train ROLE with 50 roles available (additional training information provided in Appendix A.10).

---

of cognitive representations must individually be *causally efficacious* in order for those constituents to provide an explanation of the compositionality of cognition (Fodor & McLaughlin, 1990; Fodor, 1997). That TPRs meet the challenge of explaining compositionality was argued in Smolensky (1987; 1991).

[4]We extract syntactic categories from the SCAN grammar (Lake & Baroni, 2018, Supplementary Fig. 6) by saying that two words belong to the same category if every occurrence of one could be grammatically replaced by the other. Based on our analysis in Appendix A.8, we do not replace occurrences of `and` and `after` since the presence of either of these words causes substantial changes in the roles assigned to the sequence.

Table 3: MSE loss for learned (bold) and hand-crafted role schemes on sentence embedding models.

|  | **Continuous** | **Snapped** | **Discrete** | LTR | RTL | Bi | Tree | BOW |
|---|---|---|---|---|---|---|---|---|
| InferSent | **4.05e-4** | 4.15e-4 | 5.76e-4 | 8.21e-4 | 9.70e-4 | 9.16e-4 | 7.78e-4 | 4.34e-4 |
| Skip-thought | 9.30e-5 | 9.32e-5 | 9.85e-5 | 9.91e-5 | 1.78e-3 | 3.95e-4 | 9.64e-5 | **8.87e-5** |
| SST | **5.58e-3** | 6.72e-3 | 6.48e-3 | 8.35e-3 | 9.29e-3 | 8.55e-3 | 5.99e-3 | 9.38e-3 |
| SPINN | **.139** | .151 | .147 | .184 | .189 | .181 | .178 | .176 |

As a baseline, we also train TPEs that use pre-defined role schemes (additional training information in Appendix A.9). For all of the sentence embedding models except Skip-thought, ROLE with continuous attention provides the lowest mean squared error at approximating the encoding (Table 3). The BOW (bag-of-words) role scheme represents a TPE that does not use compositional structure by assigning the same role to every filler; for each of the sentence embedding models tested except for SST, performance is within the same order of magnitude as structure-free BOW. Parikh et al. (2016) found that a bag-of-words model scores extremely well on Natural Language Inference despite having no knowledge of word order, showing that structure is not necessary to perform well on the sorts of tasks commonly used to train sentence encoders. Although not definitive, these results provide no evidence that these sentence embedding models rely on compositional representations.

## 8 FUTURE WORK

TPRs provide NNs an aspect of the systematicity of symbolic computation by disentangling fillers and roles. The learner needs to learn to process fillers — providing, essentially, *what* each input constituent contributes to the output — and to process roles — essentially, *how* these contributions are used in the output. In this work, we used ROLE to interpret the workings of a target encoder $\mathcal{E}$, and in future work, we plan to train ROLE in an end-to-end manner, either using it as the encoder itself, or using it to regularize a standard (e.g., RNN) encoder with a loss term that rewards learning compositional encodings that ROLE can approximate well. We will test whether such an explicit bias for compositionality allows networks to train faster, or with fewer parameters, and to achieve more systematic generalization. Recent work showed improvements in compositionality by separating out syntax and semantics with attention (Russin et al., 2019), and our results suggest that ROLE can also disentangle syntax and semantics.

The structured representations that TPRs provide allow us to structure the *processing* that occurs over these representations. In fully-compositional tasks, the processing of fillers and roles can be encoded (and hence in principle learned) independently. In partially-compositional tasks, the processing of fillers and roles may be approximately encoded independently, with key interactions when a task deviates from full compositionality. In this work, we showed how a hidden embedding can be factored into fillers and roles. Similarly, it is possible to factor a weight matrix into weights that process roles and weights that process fillers. An illustration of this with a simple example from the SCAN task is provided in Appendix A.11. We plan to explore whether providing networks with a bias favoring weight matrices that factor in this way can improve systematic generalization.

## 9 CONCLUSION

We have introduced ROLE, a neural network that learns to approximate the representations of an existing target neural network $\mathcal{E}$ using an explicit symbolic structure. ROLE successfully discovers symbolic structure both in models that explicitly define this structure and in an RNN without explicit structure trained on the fully-compositional SCAN task. When applied to sentence embedding models trained on partially-compositional tasks, ROLE performs better than hand-specified role schemes but still provides little evidence that the sentence encodings represent compositional structure. Uncovering the latent symbolic structure of NN representations on fully-compositional tasks is a significant step towards explaining how they can achieve the level of compositional generalization that they do, and suggests types of inductive bias to improve such generalization for partially-compositional tasks.

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

## A  Appendix

### A.1  Tensor Product Encoder (TPE) Architecture

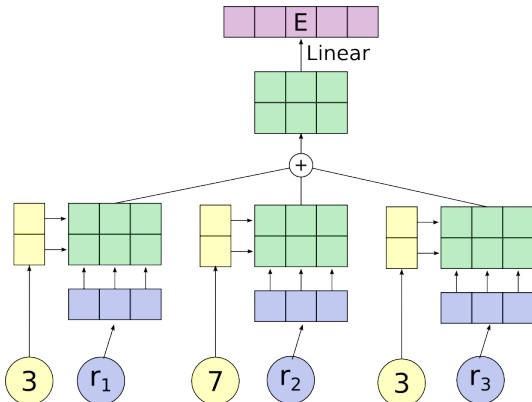

Figure 3: The Tensor Product Encoder architecture. The yellow circle is an embedding layer for the fillers, and the blue circle is an embedding layer for the roles. These two vector embeddings are combined by an outer product to produce the green matrix representing the TPR of the constituent. All of the constituents are summed together to produce the TPR of the sequence, and then a linear transformation is applied to resize the TPR to the target encoders dimensionality. ROLE replaces the role embedding layer and directly produces the blue role vector.

### A.2  Designed role schemes

We use six hand-specified role schemes as a baseline to compare the learned role schemes against. Examples of each role scheme are shown in Table 4.

1. Left-to-right (LTR): Each filler's role is its index in the sequence, counting from left to right.

2. Right-to-left (RTL): Each filler's role is its index in the sequence, counting from right to left.

3. Bidirectional (Bi): Each filler's role is a pair of indices, where the first index counts from left to right, and the second index counts from right to left.

4. Tree: Each filler's role is given by its position in a tree. This depends on a tree parsing algorithm.

5. Wickelroles (Wickel): Each filler's role is the filler before it and the filler after it. (Wickelgren, 1969)

6. Bag-of-words (BOW): Each filler is assigned the same role. The position and context of the filler is ignored.

Table 4: The assigned roles for two sequences, 3116 and 523197. Table reproduced from McCoy et al. (2019a).

| | 3 | 1 | 1 | 6 | 5 | 2 | 3 | 1 | 9 | 7 |
|---|---|---|---|---|---|---|---|---|---|---|
| Left-to-right | 0 | 1 | 2 | 3 | 0 | 1 | 2 | 3 | 4 | 5 |
| Right-to-left | 3 | 2 | 1 | 0 | 5 | 4 | 3 | 2 | 1 | 0 |
| Bidirectional | (0, 3) | (1, 2) | (2, 1) | (3, 0) | (0, 5) | (1, 4) | (2, 3) | (3, 2) | (4, 1) | (5, 0) |
| Wickelroles | #_1 | 3_1 | 1_6 | 1_# | #_2 | 5_3 | 2_1 | 3_9 | 1_7 | 9_# |
| Tree | L | RLL | RLR | RR | LL | LRLL | LRLR | LRRL | LRRR | R |
| Bag of words | $r_0$ | $r_0$ | $r_0$ | $r_0$ | $r_0$ | $r_0$ | $r_0$ | $r_0$ | $r_0$ | $r_0$ |

### A.3 TPEs TRAINED ON DIGIT SEQUENCE TASKS

We trained two TPEs end-to-end with an RNN decoder for our target networks on the digit sequence tasks. The left-to-right (LTR) TPE used a left-to-right role scheme applied to each element in the input and was connected to a unidirectional GRU decoder. The tree TPE used tree positions for each element in the input and was connected to a tree GRU decoder; here, the digit strings were parsed as a binary tree by a deterministic algorithm given in McCoy et al. (2019a, App. C). The filler and role dimensions were both 20 for the LTR TPE. The filler dimension was 20 for the Tree TPE, and the role dimension was 120. We used a hidden size of 60 for the GRU decoders. We used a patience of 2 for early stopping. The left-to-right TPE achieves 100% accuracy on the test set and the tree TPE achieves 98.62% on the test set.

### A.4 ROLE REGULARIZATION

Letting $\boldsymbol{A} = \{\boldsymbol{a}_t\}_{t=1}^T$, the regularization term applied during ROLE training is $R = \lambda(R_1 + R_2 + R_3)$, where $\lambda$ is a regularization hyperparameter and:

$$R_1(\boldsymbol{A}) = \sum_{t=1}^T \sum_{\rho=1}^{n_R} [\boldsymbol{a}_t]_\rho (1 - [\boldsymbol{a}_t]_\rho); \quad R_2(\boldsymbol{A}) = -\sum_{t=1}^T \sum_{\rho=1}^{n_R} [\boldsymbol{a}_t]_\rho^2; \quad R_3(\boldsymbol{A}) = \sum_{\rho=1}^{n_R} ([\boldsymbol{s}_{\boldsymbol{A}}]_\rho (1 - [\boldsymbol{s}_{\boldsymbol{A}}]_\rho))^2$$

Since each $\boldsymbol{a}_t$ results from a softmax, its elements are positive and sum to 1. Thus the factors in $R_1(\boldsymbol{A})$ are all non-negative, so $R_1$ assumes its minimal value of 0 when each $\boldsymbol{a}_t$ has binary elements; since these elements must sum to 1, such an $\boldsymbol{a}_t$ must be one-hot. $R_2(\boldsymbol{A})$ is also minimized when each $\boldsymbol{a}_t$ is one-hot because when a vector's $L^1$ norm is 1, its $L^2$ norm is maximized when it is one-hot. Although each of these terms individually favor one-hot vectors, empirically we find that using both terms helps the training process. In a discrete symbolic structure, each position can hold at most one symbol, and the final term $R_3$ in ROLE's regularizer $R$ is designed to encourage this. In the vector $\boldsymbol{s}_{\boldsymbol{A}} = \sum_{t=1}^T \boldsymbol{a}_t$, the $\rho^{\text{th}}$ element is the total attention weight, over all symbols in the string, assigned to the $\rho^{\text{th}}$ role: in the discrete case, this must be 0 (if no symbol is assigned this role) or 1 (if a single symbol is assigned this role). Thus $R_3$ is minimized when all elements of $\boldsymbol{s}$ are 0 or 1 ($R_3$ is similar to $R_1$, but with squared terms since we are no longer assured each element is at most 1). It is important to normalize each role embedding in the role matrix **R** so that small attention weights have correspondingly small impacts on the weighted-sum role embedding.

### A.5 ROLE TRAINED ON DIGIT SEQUENCE TASKS

Once the TPEs in Sec. A.3 were trained, we extracted the hidden embedding for each item in the training, dev, and test sets.

For both ROLE models trained on the digit sequence task, we used a bidirectional 2-layer LSTM (Hochreiter & Schmidhuber, 1997) with filler dimension of 20, and regularization constant $\lambda = 1$. For training, we used the ADAM (Kingma & Ba, 2015) optimizer with a learning rate of .001, batch size 32, and an early stopping patience of 10. The ROLE model trained on the LTR TPE was given 20 roles each of dimension 20. The ROLE model trained on the Tree TPE was given 120 roles each of dimension 120.

### A.6 RNN TRAINED ON SCAN

To train the standard RNN on SCAN, we ran a limited hyperparameter search similar to the procedure in Lake & Baroni (2018). Since our goal was to produce a single embedding that captured the entire input sequence, we fixed the architecture to GRU with a single hidden layer. We did not train models with attention, since we wanted to investigate whether a standard RNN could capture compositionality. The remaining hyperparameters were hidden dimension and dropout. We ran a search over the hidden dimension sizes of 50, 100, 200, and 400 as well as dropout with a value of 0, .1, and .5 applied to the word embeddings and recurrent layer. Each network was trained with the ADAM optimizer (Kingma & Ba, 2015) and a learning rate of .001 for 100,000 steps with a batch-size of 1. The best performing network had a hidden dimension or 100 and dropout of .1.

## A.7 ROLE TRAINED ON SCAN

For the ROLE models trained to approximate the GRU encoder trained on SCAN, we used a filler dimension of 100, a role dimension of 50 with 50 roles available. For training, we used the ADAM (Kingma & Ba, 2015) optimizer with a learning rate of .001, batch size 32, and an early stopping patience of 10. The role assignment module used a bidirectional 2-layer LSTM (Hochreiter & Schmidhuber, 1997). We performed a hyperparameter search over the regularization coefficient $\lambda$ using the values in the set [.1, .02, .01]. The best performing value was .02, and we used this model in our analysis.

## A.8 SCAN ROLE ANALYSIS

The algorithm below characterizes our post-hoc interpretation of which roles the Role Learner will assign to elements of the input to the SCAN model. This algorithm was created by hand based on an analysis of the Role Learner's outputs for the elements of the SCAN training set. The algorithm works equally well on examples in the training set and the test set; on both datasets, it exactly matches the roles chosen by the Role Learner for 98.7% of sequences (20,642 out of 20,910).[5]

### A.8.1 A ROLE-ASSIGNMENT ALGORITHM IMPLICITLY LEARNED BY THE SCAN SEQ2SEQ ENCODER

The input sequences have three basic types that are relevant to determining the role assignment: sequences that contain *and* (e.g., *jump around left and walk thrice*), sequences that contain *after* (e.g., *jump around left after walk thrice*), and sequences without *and* or *after* (e.g., *turn opposite right thrice*). Within commands containing *and* or *after*, it is convenient to break the command down into the command before the connecting word and the command after it; for example, in the command *jump around left after walk thrice*, these two components would be *jump around left* and *walk thrice*.

- Sequence with *and*:
    - Elements of the command before *and*:
        * Last word: 28
        * First word (if not also last word): 46
        * *opposite* if the command ends with *thrice*: 22
        * Direction word between *opposite* and *thrice*: 2
        * *opposite* if the command does not end with *thrice*: 2
        * Direction word after *opposite* but not before *thrice*: 4
        * *around*: 22
        * Direction word after *around*: 2
        * Direction word between an action word and *twice* or *thrice*: 2
    - Elements of the command before *and*:
        * First word: 11
        * Last word (if not also the first word): 36
        * Second-to-last word (if not also the first word): 3
        * Second of four words: 24
    - *and*: 30
- Sequence with *after*:
    - Elements of the command before *after*:
        * Last word: 8
        * Second-to-last word: 36
        * First word (if not the last or second-to-last word): 11
        * Second word (if not the last or second-to-last word): 3
    - Elements of the command after *after*:
        * Last word: 46

---

[5]This figure of 98.7% is so constant across datasets presumably because the synthetic nature of the SCAN dataset means that any reasonably-sized sample from it will be similarly representative of the entire dataset.

* Second-to-last word: 4
* First word if the command ends with *around right*: 4
* First word if the command ends with *thrice* and contains a rotation: 10
* First word if the command does not end with *around right* and does not contain both *thrice* and a rotation: 17
* Second word if the command ends with *thrice*: 17
* Second word if the command does not end with *thrice*: 10

  - *after*: 17 if no other word has role 17 or if the command after *after* ends with *around left*; 43 otherwise

- Sequence without *and* or *after*:
  - Action word directly before a cardinality: 4
  - Action word before, but not directly before, a cardinality: 34
  - *thrice* directly after an action word: 2
  - *twice* directly after an action word: 2
  - *opposite* in a sequence ending with *twice*: 8
  - *opposite* in a sequence ending with *thrice*: 34
  - *around* in a sequence ending with a cardinality: 22
  - Direction word directly before a cardinality: 2
  - Action word in a sequence without a cardinality: 46
  - *opposite* in a sequence without a cardinality: 2
  - Direction after *opposite* in a sequence without a cardinality: 26
  - *around* in a sequence without a cardinality: 3
  - Direction after *around* in a sequence without a cardinality: 22
  - Direction directly after an action in a sequence without a cardinality: 22

To show how this works with an example, consider the input *jump around left after walk thrice*. The command before *after* is *jump around left*. *left*, as the last word, is given role 8. *around*, as the second-to-last word, gets role 36. *jump*, as a first word that is not also the last or second-to-last word gets role 11. The command after *after* is *walk thrice*. *thrice*, as the last word, gets role 46. *walk*, as the second-to-last word, gets role 4. Finally, *after* gets role 17 because no other elements have been assigned role 17 yet. These predicted outputs match those given by the Role Learner.

### A.8.2 DISCUSSION OF THE ALGORITHM

We offer several observations about this algorithm.

1. This algorithm may seem convoluted, but a few observations can illuminate how the roles assigned by such an algorithm support success on the SCAN task. First, a sequence will contain role 30 if and only if it contains *and*, and it will contain role 17 if and only if it contains *after*. Thus, by implicitly checking for the presence of these two roles (regardless of the fillers bound to them), the decoder can tell whether the output involves one or two basic commands, where the presence of *and* or *after* leads to two basic commands and the absence of both leads to one basic command. Moreover, if there are two basic commands, whether it is role 17 or role 30 that is present can tell the decoder whether the input order of these commands also corresponds to their output order (when it is *and* in play, i.e., role 30), or if the input order is reversed (when it is *after* in play, i.e., role 17).

   With these basic structural facts established, the decoder can begin to decode the specific commands. For example, if the input is a sequence with *after*, it can begin with the command after *after*, which it can decode by checking which fillers are bound to the relevant roles for that type of command.

   It may seem odd that so many of the roles are based on position (e.g., "first word" and "second-to-last word"), rather than more functionally-relevant categories such as "direction word." However, this approach may actually be more efficient: Each command consists of a single mandatory element (namely, an action word such as *walk* or *jump*) followed by several optional modifiers (namely, rotation words, direction words, and cardinalities).

Because most of the word categories are optional, it might be inefficient to check for the presence of, e.g., a cardinality, since many sequences will not have one. By contrast, every sequence will have a last word, and checking the identity of the last word provides much functionally-relevant information: if that word is not a cardinality, then the decoder knows that there is no cardinality present in the command (because if there were, it would be the last word); and if it is a cardinality, then that is important to know, because the presence of *twice* or *thrice* can dramatically affect the shape of the output sequence. In this light, it is unsurprising that the SCAN encoder has implicitly learned several different roles that essentially mean the last element of a particular subcommand.

2. The algorithm does not constitute a simple, transparent role scheme. But its job is to describe the representations that the original network produces, and we have no a priori expectation about how complex that process may be. The role-assignment algorithm implicitly learned by ROLE is interpretable locally (each line is readily expressible in simple English), but not intuitively transparent globally. We see this as a positive result, in two respects.

    First, it shows why ROLE is crucial: no human-generated role scheme would provide a good approximation to this algorithm. Such an algorithm can only be identified because ROLE is able to use gradient descent to find role schemes far more complex than any we would hypothesize intuitively. This enables us to analyze networks far more complex than we could analyze previously, being necessarily limited to hand-designed role schemes based on human intuitions about how to perform the task.

    Second, when future work illuminates the computation in the original SCAN GRU seq2seq decoder, the baroqueness of the role-assignment algorithm that ROLE has shown to be implicit in the seq2seq encoder can potentially explain certain limitations in the original model, which is known to suffer from severe failures of systematic generalization outside the training distribution (Lake and Baroni, 2018). It is reasonable to hypothesize that systematic generalization requires that the encoder learn an implicit role scheme that is relatively simple and highly compositional. Future proposals for improving the systematic generalization of models on SCAN can be examined using ROLE to test the hypothesis that greater systematicity requires greater compositional simplicity in the role scheme implicitly learned by the encoder.

3. While the role-assignment algorithm of A.8.1 may not be simple, from a certain perspective, it is quite surprising that it is not far more complex. Although ROLE is provided 50 roles to learn to deploy as it likes, it only chooses to use 16 of them (only 16 are ever selected as the $\arg\max(\boldsymbol{a}_t)$; see Sec. 6.1). Furthermore, the SCAN grammar generates 20,910 input sequences, containing a total of 151,688 words (an average of 7.25 words per input). This means that, if one were to generate a series of conditional statements to determine which role is assigned to each word in every context, this could in theory require up to 151,688 conditionals (e.g., "if the filler is 'jump' in the context 'walk thrice after ___ opposite left', then assign role 17"). However, our algorithm involves just 47 conditionals. This reduction helps explain how the model performs so well on the test set: If it used many more of the 151,688 possible conditional rules, it would completely overfit the training examples in a way that would be unlikely to generalize. The 47-conditional algorithm we found is more likely to generalize by abstracting over many details of the context.

4. Were it not for ROLE's ability to characterize the representations generated by the original encoder in terms of implicit roles, providing an equally complete and accurate interpretation of those representations would necessarily require identifying the conditions determining the activation level of each of the 100 neurons hosting those representations. It seems to us grossly overly optimistic to estimate that each neuron's activation level in the representation of a given input could be characterized by a property of the input statable in, say, two lines of roughly 20 words/symbols; yet even then, the algorithm would require 200 lines, whereas the algorithm in A.8.1 requires 47 lines of that scale. Thus, by even such a crude estimate of the degree of complexity expected for an algorithm describing the representations in terms of neuron activities, the algorithm we find, stated over roles, is 4 times simpler.

## A.9 TPEs TRAINED ON SENTENCE EMBEDDING MODELS

For each sentence embedding model, we trained three randomly initialized TPEs for each role scheme and selected the best performing one as measured by the lowest MSE. For each TPE, we used the

original filler embedding from the sentence embedding model. This filler dimensionality is 25 for SST, 300 for SPINN and InferSent, and 620 for Skipthought. We applied a linear transformation to the pre-trained filler embedding where the input size is the dimensionality of the pre-trained embedding and the output size is also the dimensionality of the pre-trained embedding. This linearly transformed embedding is used as the filler vector in the filler-role binding in the TPE. For each TPE, we use a role dimension of 50. Training was done with a batch size of 32 using the ADAM optimizer with a learning rate of .001.

To generate tree roles from the English sentences, we used the constituency parser released in version 3.9.1 of Stanford CoreNLP (Klein & Manning, 2003).

### A.10 ROLE TRAINED ON SENTENCE EMBEDDING MODELS

For each sentence embedding model, we trained three randomly initialized ROLE models and selected the best performing one as measured by the lowest MSE. We used the original filler embedding from the sentence embedding model (25 for SST, 300 for SPINN and InferSent, and 620 for Skipthought). We applied a linear transformation to the pre-trained filler embedding where the input size is the dimensionality of the pre-trained embedding and the output size is also the dimensionality of the pre-trained embedding. This linearly transformed embedding is used as the filler vector in the filler-role binding in the TPE. We also applied a similar linear transformation to the pre-trained filler embedding before input to the role learner LSTM. For each ROLE model, we provide up to 50 roles with a role dimension of 50. Training was done with a batch size of 32 using the ADAM optimizer with a learning rate of .001. We performed a hyperparameter search over the regularization coefficient $\lambda$ using the values in the set $\{1, 0.1, 0.01, 0.001, 0.0001\}$. For SST, SPINN, InferSent and SST, respectively, the best performing network used $\lambda = 0.001, 0.01, 0.001, 0.1$.

### A.11 FACTORING WEIGHTS TO PROCESS FILLERS AND ROLES INDEPENDENTLY IN SCAN

At the end of Sec. 8 we remarked: "the structured representation bias that TPRs provide allows us to also provide biases to structure the processing that occurs over these representations. In fully-compositional tasks, the processing of fillers and roles can be encoded (and hence in principle learned) independently. ... In this work, we showed how a hidden embedding can be factored into fillers and roles. Similarly, it is possible to factor a weight matrix into weights that process roles and weights that process fillers." Here we exemplify this with SCAN.

One of the compositional rules for the mapping $\varphi$ defined by the SCAN task is: $\varphi(x \text{ twice}) = \varphi(x) \, \varphi(x)$; for example, $\varphi(\text{walk twice}) = \varphi(\text{walk}) \, \varphi(\text{walk}) = \text{WALK WALK}$. For the purposes of illustration, suppose that, given the input string $x \text{ twice}$, a NN encoder produces a representation that is approximated by the TPR of a single-constituent structure in which the filler $x$ is assigned the role "argument of $\text{twice}$": $x : r_{2\text{ce-arg}}$. So the TPR encoding of $\text{jump twice}$ is $\mathbf{e}(\text{jump twice}) = \mathbf{e}(\text{jump} : r_{2\text{ce-arg}}) = \mathbf{e}_\text{F}(\text{jump}) \otimes \mathbf{e}_\text{R}(r_{2\text{ce-arg}})$, where $\mathbf{e}_\text{F}, \mathbf{e}_\text{R}$ are the embedding functions for fillers and roles, respectively. Let us also suppose that the output string is encoded as a TPR with positional roles $R_i$, so that $\text{WALK LOOK}$ has filler:role bindings $\{\text{WALK} : R_1, \text{LOOK} : R_2\}$ and so has TPR $\mathbf{e}\left(\{\text{WALK} : R_1, \text{LOOK} : R_2\}\right) = \mathbf{e}_\text{F}(\text{WALK}) \otimes \mathbf{e}_\text{R}(R_1) + \mathbf{e}_\text{F}(\text{LOOK}) \otimes \mathbf{e}_\text{R}(R_2)$.

So to generalize correctly to the input $\text{jump twice}$, producing output $\text{JUMP JUMP}$, the system needs to learn two things: how to map the filler — $\varphi_\text{F}(\text{jump}) = \text{JUMP}$ — and how to map the role — $\varphi_\text{R}(r_{2\text{ce-arg}}) = \{R_1, R_2\}$. The desired mapping is $\varphi : \mathbf{e}(\text{jump} : r_{2\text{ce-arg}}) \mapsto \mathbf{e}(\text{JUMP JUMP})$, that is, $\varphi : \mathbf{e}_\text{F}(\text{jump}) \otimes \mathbf{e}_\text{R}(r_{2\text{ce-arg}}) \mapsto \mathbf{e}_\text{F}(\text{JUMP}) \otimes \mathbf{e}_\text{R}(R_1) + \mathbf{e}_\text{F}(\text{JUMP}) \otimes \mathbf{e}_\text{R}(R_2)$. We now show that this can be accomplished through the separate filler- and role-mappings $\varphi_\text{F} : \mathbf{e}_\text{F}(\text{jump}) \mapsto \mathbf{e}_\text{F}(\text{JUMP})$ and $\varphi_\text{R} : \mathbf{e}_\text{R}(r_{2\text{ce-arg}}) \mapsto \mathbf{e}_\text{R}(R_1) + \mathbf{e}_\text{R}(R_2)$.

To do this, we show that if $\varphi_\text{F}, \varphi_\text{R}$ are respectively computed over embedding vectors through weight matrices $\mathbb{W}_\text{F}, \mathbb{W}_\text{R}$ in a NN, then the weight tensor $\mathbb{W} = \mathbb{W}_\text{F} \otimes \mathbb{W}_\text{R}$ will correctly map the embedding of $\text{jump twice}$ to the embedding of $\text{JUMP JUMP}$. The tensor product of weight matrices is defined as $[\mathbb{W}_\text{F} \otimes \mathbb{W}_\text{R}]_{ik}^{jl} = [\mathbb{W}_\text{F}]_i^j [\mathbb{W}_\text{R}]_k^l$ which entails that $(\mathbb{W}_\text{F} \otimes \mathbb{W}_\text{R})[\mathbf{f} \otimes \mathbf{r}] = [\mathbb{W}_\text{F}\mathbf{f}] \otimes [\mathbb{W}_\text{R}\mathbf{r}]$. Therefore $\mathbb{W}[\mathbf{e}(\text{jump twice})] = \mathbb{W}[\mathbf{e}(\text{jump} : r_{2\text{ce-arg}})] = \mathbb{W}[\mathbf{e}_\text{F}(\text{jump}) \otimes \mathbf{e}_\text{R}(r_{2\text{ce-arg}})] = [\mathbb{W}_\text{F}\mathbf{e}_\text{F}(\text{jump})] \otimes [\mathbb{W}_\text{R}\mathbf{e}_\text{R}(r_{2\text{ce-arg}})] = \mathbf{e}_\text{F}(\text{JUMP}) \otimes [\mathbf{e}_\text{R}(R_1) + \mathbf{e}_\text{R}(R_2)] = \mathbf{e}_\text{F}(\text{JUMP}) \otimes \mathbf{e}_\text{R}(R_1) + \mathbf{e}_\text{F}(\text{JUMP}) \otimes \mathbf{e}_\text{R}(R_2) = \mathbf{e}(\text{JUMP JUMP})$, as desired.

This suggests that greater compositional generalization might be achieved by future models that are explicitly biased to utilize TPRs as internal representations and explicitly biased to factor their processing weights into those that process fillers (independently of their roles) and those that process roles (independently of their fillers), as in the SCAN example with $\mathbb{W} = \mathbb{W}_F \otimes \mathbb{W}_R$ above.

