# OpenReview forum: "Discovering the compositional structure of vector representations with Role Learning Networks"
_ICLR.cc/2020/Conference — Reject_

### Official Review · AnonReviewer3 · 2019-10-23
**Official Blind Review #3**

**Rating:** 6

**Review:**

In this paper, the authors study the problem of understanding the compositional generalization abilities of NNs. A new type of NN, called ROLE, is proposed to learns to approximate the representations of a target NN E by learning a symbolic constituent structure and an embedding of that structure intoE’s representational vector space. A number of tasks are conducted, including a simple fully-compositional task, a SCAN task, a partially-compositional NLP task. Multiple tasks. The experiment results show that the proposed approach can help to understand how NNs achieve strong generalization on partially-compositional tasks, and good performance on fully-compositional task.
Pros:
1. This work studies an important problem of fundamental AI.
2. Authors conduct experiments on multiple task to evaluate the effectiveness of the proposed technique and show how it helps to understand the generalization of NNs.
3. The overall paper is well written, except for some typos, e.g. in page 5, Table 1 Should "1.0" be "100%"? Should "0.828 be ''82.80".
Cons:
1. Regarding the Q2 raised in the paper, only simple suggestions are given. It is suggested to evaluate its effectiveness on some real experiments.

**Experience Assessment:**

I do not know much about this area.

**Review Assessment: Checking Correctness Of Derivations And Theory:**

I assessed the sensibility of the derivations and theory.

**Review Assessment: Checking Correctness Of Experiments:**

I carefully checked the experiments.

**Review Assessment: Thoroughness In Paper Reading:**

I read the paper at least twice and used my best judgement in assessing the paper.

---

> ### Author Response · Authors · 2019-11-13
> **Author response**
>
> Thank you for your helpful comments, which we incorporated into the revised paper we have  uploaded to OpenReview.  This contains a new Section A.8.2 of the Appendix, which makes several points of general importance in its substantial discussion of the algorithm ROLE implicitly uses to assign roles.
>
> Please feel free to let us know if you spot any remaining unaddressed typos.
>
> As for Table 1, the final column is the V-measure score, which ranges from 0 to 1. This does have an interpretation similar to a percentage, but the literature tends to keep the value in this range of 0 to 1.
>
>
> Because you raise a point also made by Reviewer 2, let us repeat our response here. Regarding our suggestions for how improved understanding of the compositional structure of RNN representations might lead to improved future architectures, we agree that these were brief and tentative, so we have rephrased Sec. 1 to focus the overview on the main issue addressed, a question of understanding: “How do NNs achieve such strong generalization on partially-compositional tasks, and good performance on fully-compositional tasks?“ We  have also expanded our concrete proposal for a ‘factorization bias’ into a new Section A.11 of the Appendix, and have inserted additional text in Sec. 8 where we make more substantial suggestions for improving future models: “In this work, we used ROLE to interpret the workings of a target encoder $\mathcal{E}$, and in future work, we plan to train ROLE in an end-to-end manner, either using it as the encoder itself, or using it to regularize a standard (e.g., RNN) encoder with a loss term that rewards learning compositional encodings, operationalized as encodings that ROLE can approximate well. We will test whether such an explicit bias for compositionality allows networks to train faster, or with fewer parameters, and to achieve more systematic generalization. Recent work showed improvements in compositionality by separating out syntax and semantics with attention (Russin et al., 2019), and our results suggest that ROLE can also disentangle syntax and semantics.” [p. 8]
>
> Thank you for your review, which we believe has helped us to improve the paper considerably. If you should have any further comments, they would be most welcome.

---

### Official Review · AnonReviewer2 · 2019-10-23
**Official Blind Review #2**

**Rating:** 3

**Review:**


This work focuses on the understanding of Deep NNs and of their generalization ability for compositional tasks and especially for partially composition tasks such as one encounters in natural language processing.

The core idea is to design and to learn a neural model (the ROLE network) able to analyse a target neural network (e.g. an encoder in an seq2seq architecture) by identifying the symbolic structures the target network manipulates (the symbols and their roles) in its representations and the compositional rules it has learned on these. The main motivations are the understanding of what is learnt by a such neural network and getting ideas on which architecture to choose for improving generalization on compositional tasks.

The experimental study shows that ROLE ’s results are relevant when dealing with a fully compositional model by design. This is a kind of sanity check, that ROLE may uncover the ground truth from the data, while no prior information is provided on the nature of the roles.
The experiments on the SCAN dataset concern a standard RNN model learned from data. I am not sure of the nature of RNN used here. How many layers, which activation function ? The experimental section also provides insights on what is learned by the ROLE model and on how its (compositional) representations match the ones learned by the RNN.

As far as in understand, beyond the understanding of the learned representations in RNNs, the paper motivates the work with the expectation that the gained knowledge might be useful for designing better neural nets, with improved generalization ability. Yet i don’t  see clearly what could be done on this line.

**Experience Assessment:**

I do not know much about this area.

**Review Assessment: Checking Correctness Of Derivations And Theory:**

I assessed the sensibility of the derivations and theory.

**Review Assessment: Checking Correctness Of Experiments:**

I assessed the sensibility of the experiments.

**Review Assessment: Thoroughness In Paper Reading:**

I read the paper at least twice and used my best judgement in assessing the paper.

---

> ### Author Response · Authors · 2019-11-13
> **Author response**
>
> Thank you for your suggestions, which we believe have led to an improved paper (revision has been uploaded to OpenReview). In addition to the revisions mentioned below, let us point out a new Section of the Appendix, A.8.2, which contains considerable discussion of the algorithm ROLE implicitly uses to assign roles; we think there are several important points there.
>
> Regarding the standard RNN used on SCAN, the current revision adds an additional section to the appendix (A.6) where we discuss this in some detail. As for the activation function, we used a standard GRU cell, which includes both sigmoid and tanh activation functions.
>
> As for our suggestions for how improved understanding of the compositional structure of RNN representations might lead to improved future architectures, we agree that these were brief and tentative, so we have rephrased Sec. 1 to focus the overview on the main issue addressed, a question of understanding: “How do NNs achieve such strong generalization on partially-compositional tasks, and good performance on fully-compositional tasks?“ We  have also expanded our concrete proposal for a ‘factorization bias’ into a new Section A.11 of the Appendix, and have inserted additional text in Sec. 8 where we make more substantial suggestions for improving future models: “In this work, we used ROLE to interpret the workings of a target encoder $\mathcal{E}$, and in future work, we plan to train ROLE in an end-to-end manner, either using it as the encoder itself, or using it to regularize a standard (e.g., RNN) encoder with a loss term that rewards learning compositional encodings, operationalized as encodings that ROLE can approximate well. We will test whether such an explicit bias for compositionality allows networks to train faster, or with fewer parameters, and to achieve more systematic generalization. Recent work showed improvements in compositionality by separating out syntax and semantics with attention (Russin et al., 2019), and our results suggest that ROLE can also disentangle syntax and semantics.” [p. 8]
>
> Thanks again for your comments, which we believe have allowed us to strengthen the paper appreciably. Any further comments you might have are most welcome.

---

### Official Review · AnonReviewer1 · 2019-10-28
**Official Blind Review #1**

**Rating:** 6

**Review:**

The paper introduces an approach, called ROLE, that extract symbolic structure from seq2seq networks.  It also provides an interpretable symbolic structure and examines the causal information in the symbolic structure.

The approach is inspired by the Tensor Product Encoder architecture.

The scan role analysis part seemed the most hand-wavy with lots of positions in A.7.

None of the accuracy results have variances attached to them.

I am not an expert on this topic (hence the weak accept), but I liked the paper.



**Experience Assessment:**

I have read many papers in this area.

**Review Assessment: Checking Correctness Of Derivations And Theory:**

I assessed the sensibility of the derivations and theory.

**Review Assessment: Checking Correctness Of Experiments:**

I assessed the sensibility of the experiments.

**Review Assessment: Thoroughness In Paper Reading:**

I read the paper at least twice and used my best judgement in assessing the paper.

---

> ### Author Response · Authors · 2019-11-13
> **Author response**
>
> Thank you for your feedback! We have done our best to incorporate the suggestions from you and the other reviewers into the revised version of the paper now uploaded to OpenReview. Your comments led to considerable revisions of the paper, as the length of this comment attests. We hope you will agree that the paper is notably stronger as a result.
>
> Following your suggestion, we are computing the variance in our results; we expect that this will be incorporated into a final revision that we will upload on Friday Nov 15.
>
> Regarding the SCAN role analysis, there are 3 facets which may contribute to your concern that we’d like to comment on.
>
> 1. As mentioned in the paper, the role-assignment algorithm now in Appendix A.8.1 was indeed hand-generated, as were the admittedly speculative comments in Sec. 6.2 about how such a role scheme might contribute to the strong performance of the original GRU seq2seq SCAN network being analyzed by ROLE. In our view, however, interpretation of neural nets has an inevitable human component, as the goal is to connect the formal properties of the network to (informal) human understanding. We see these human-generated observations as hypotheses to test; although the generation of hypotheses may not be formal, testing them can be. Formally testing the hypothesized algorithm in Sec. A.8.1 is straightforward, and as reported in Sec. A.8.1, the result is 98.7% accuracy on the test set, which was not consulted when generating the hypothesized algorithm from the training set alone: the algorithm does not over-fit the data used to generate it, but robustly generalizes to held-out data. Thus, while the hypothesized algorithm was informally generated, it received strong formal verification: the best outcome possible, in our view.
>
> 2. The revised paper makes the following argument in Appendix A.8.2 (p. 17): “ The algorithm in A.8 does not constitute a simple, transparent role scheme. But its job is to describe the representations that the original network produces, and we have no a priori expectation about how complex that process may be. The role-assignment algorithm implicitly learned by ROLE is interpretable locally (each line is readily expressible in English), but not intuitively transparent globally. We see this as a positive result, in two respects.
>
> First, it shows why ROLE is crucial: no human-generated role scheme would provide a good approximation to this algorithm; it can only be identified because ROLE is able to use gradient descent to find role schemes far more complex than any we would hypothesize intuitively. This enables us to analyze networks far more complex than we could analyze previously when we were limited to hand-designed role schemes based on human intuitions about how to perform the task.
> Second, when future work illuminates the computation in the original SCAN GRU seq2seq decoder, the baroqueness of the role-assignment algorithm that ROLE has shown to be implicit in the seq2seq encoder can potentially explain certain limitations in the original model, which is known to suffer from severe failures of systematic generalization outside the training distribution (Lake and Baroni, 2018). It is reasonable to hypothesize that systematic generalization requires that the encoder learn an implicit role scheme that is relatively simple and highly compositional. Future proposals for improving the systematic generalization of models on SCAN can be examined using ROLE to test the hypothesis that greater systematicity requires greater compositional simplicity in the role scheme implicitly learned by the encoder.”
>
> 3. While the role-assignment algorithm of A.8.1 may not be simple, it is quite surprising that it is not far more complex. Although ROLE is provided 50 roles to learn to deploy as it likes, it only chooses to use 16 of them (only 16 are ever selected as the argmax($\mathbf{a}_t$); see Sec. 6.1). And as stated in the revised paper in Sec. A.8.2 (p. 17): “The SCAN grammar generates 20,910 input sequences, containing a total of 151,688 words (an average of 7.25 words per input). This means that, if one were to generate a series of conditional statements to determine which role is assigned to each word in every context, this could in theory require up to 151,688 conditionals (e.g., “if the filler is ‘jump’ in the context ‘walk thrice after ____ opposite left’, then assign role 17”). However, our algorithm involves just 47 conditionals. This reduction helps explain how the model performs so well on the test set:If it used many more of the 151,688 possible conditional rules, it would completely overfit the training examples in a way that would be unlikely to generalize. The 47-conditional algorithm we found is more likely to generalize by abstracting over many details of the context.”
>
> We thank you again for your comments, which have led us to improve the paper significantly. We welcome any further comments you may have.

---

> ### Author Response · Authors · 2019-11-15
> **Author response**
>
> Hello, the current revision includes the mean accuracy across three runs, as well standard deviations.

---

### Decision · Program_Chairs · 2019-12-19

**Decision:**

Reject

**Comment:**

This work builds directly on McCoy et al. (2019a) and add a RNN that can replace what was human generated hypotheses to the role schemes. The final goal of ROLE is to analyze a network by identifying ‘symbolic structure’. The authors conduct sanity check by conducting experiments with ground truth, and extend the work further to apply it to a complex model. I wonder under what definition of ‘interpretable’ authors have in mind with the final output (figure 2) - the output is very complex. It remains questionable if this will give some ‘insight’ or how would humans parse this info such that it is ‘useful’ for them in some way.

Overall, though this is a good paper, due to the number of strong papers this year, it cannot be accepted at this time. We hope the comments given by reviewers can help improve a future version.